# Effects of 12-Week Ingestion of Yogurt Containing *Lactobacillus plantarum* OLL2712 on Glucose Metabolism and Chronic Inflammation in Prediabetic Adults: A Randomized Placebo-Controlled Trial

**DOI:** 10.3390/nu12020374

**Published:** 2020-01-31

**Authors:** Takayuki Toshimitsu, Ayako Gotou, Toshihiro Sashihara, Satoshi Hachimura, Nobuhiko Shioya, Satoru Suzuki, Yukio Asami

**Affiliations:** 1Applied Microbiology Research Department, Food Microbiology Research Laboratories, Division of Research and Development, Meiji Co., Ltd., Hachiouji, Tokyo 192-0919, Japan; ayako.gotou@meiji.com (A.G.); toshihiro.sashihara@meiji.com (T.S.); yukio.asami@meiji.com (Y.A.); 2Research Center for Food Safety, Graduate School of Agricultural and Life Sciences, The University of Tokyo, Bunkyo-ku, Tokyo 113-8657, Japan; ahachi@mail.ecc.u-tokyo.ac.jp; 3Statistical Analysis Department, KSO Corporation, Minato-ku, Tokyo 105-0023, Japan; shioya@kso.co.jp; 4Shinagawa Season Terrace Health Care Clinic, Minato-ku, Tokyo 108-0075, Japan; satoru_suzuki@sempos.or.jp

**Keywords:** fasting blood glucose, glycoalbmin, hemoglobin A1c, high sensitivity C-reactive protein, interleukin-6, HOMA-IR

## Abstract

The ingestion of *Lactobacillus plantarum* OLL2712 (OLL2712) cells improved glucose metabolism by suppressing chronic inflammation in mouse models and in a preliminary clinical study. We aimed to clarify the effect of OLL2712 on glucose metabolism and chronic inflammation for healthy adults. Prediabetic adults (*n* = 130, age range: 20–64 years) were randomly assigned to either the placebo or OLL2712 groups (*n* = 65 each) and were administered conventional yogurt or yogurt containing more than 5 × 10^9^ heat-treated OLL2712 cells, respectively, daily for 12 weeks. Reduced HbA1c levels after 12 weeks of treatment were observed in both groups compared to those at baseline; however, the 12-week reduction of HbA1c levels was significantly greater in the OLL2712 group than in the placebo group. Increased chronic inflammation marker levels and insulin-resistant index (HOMA-IR) occurred in the placebo group but not in the OLL2712 group. Fasting blood glucose (FBG) levels did not change significantly in both groups; however, in subgroup analyses including participants with higher FBG levels, FBG levels were significantly reduced only in the OLL2712 group compared to baseline. These results suggest that OLL2712 cell ingestion can reduce HbA1c levels and can prevent the aggravation of chronic inflammation and insulin resistance.

## 1. Introduction

The prevalence of type-2 diabetes mellitus (T2DM) is prominently increasing worldwide; T2DM deteriorates health and increases medical expenses. Aging, sedentary lifestyle, unhealthy dietary behaviors, and obesity are principal risk factors for T2DM [1]. While nutritional counseling and regular exercises are effective for the improvement and prevention of T2DM, ingestion of certain kinds of functional foods also has promising effects [2]. In particular, some recent clinical trials have suggested the usefulness of lactic acid bacteria (LAB) against T2DM [3]. However, the effect of LAB in improving glucose metabolism-related parameters such as hemoglobin A1c (HbA1c) remains controversial [4], and the characteristics of LAB that make it effective for ameliorating or preventing T2DM have not been clarified.

Chronic low-grade inflammation is considered a primal cause of the development of metabolic disorders, including T2DM [5,6]. While reducing obesity is effective in reducing chronic inflammation and in preventing T2DM, weight-loss strategies are difficult for overweight individuals to achieve and maintain in the long term [7,8]. Certain LAB strains are known to have anti-inflammatory activities both in vitro and in vivo [9]. Anti-inflammatory activity is deduced to be an important mechanism of action for LAB in preventing or treating metabolic disorders [9]. The alleviation of chronic inflammation by ingesting LAB might prevent insulin resistance and might improve glucose metabolism in the adipose, muscle, and liver tissues [10]; however, it remained undetermined in previous clinical trials [11,12,13]. Andreasen et al. reported that intake of *Lactobacillus acidophilus* NCFM for 4 weeks preserved insulin sensitivity compared to placebo but did not affect the systemic inflammatory response in T2DM patients [11]. Recently, Tanaka et al. reported that heat-killed *Lactobacillus plantarum* L-137 improved inflammation and lipid metabolism in overweight healthy adults [12]. They showed that serum levels of aspartate transaminase (AST) and alanine aminotransferase (ALT), biomarkers of hepatic inflammation, were decreased compared to placebo; however, the reduction in high-sensitivity C-reactive protein (hsCRP) or pro-inflammatory cytokines, the most important biomarkers of systemic chronic inflammation, was not found.

Previously, we selected *Lactobacillus plantarum* OLL2712 (OLL2712) as an optimal anti-inflammatory LAB strain among hundreds in our LAB library [14]. The administration of heat-treated OLL2712 cells alleviated chronic inflammation by suppressing pro-inflammatory cytokine levels in visceral adipose tissue and the serum and improved hyperglycemia in mouse models with obesity and diabetes [14,15,16]. Additionally, we conducted an open-label, single-arm pilot study and confirmed the possibility that the 12-week ingestion of heat-treated OLL2712 cells would improve insulin resistance and glucose metabolism by suppressing serum pro-inflammatory cytokine levels in human prediabetic participants [17]. 

In the present study, we conducted a randomized, double-blind, placebo-controlled, parallel-group trial to examine whether the 12-week ingestion of a test yogurt containing heat-treated OLL2712 cells is effective in improving glucose metabolism-related parameters in human prediabetic participants. Yogurt is a worldwide, affordable dairy food with a low glycemic index, contains various nutrients [18], and can be easily ingested as a daily meal. The economic impact on the global healthcare burden of this common food may be substantial if the hyperglycemic condition could be managed in prediabetic participants, and their quality of life could be maintained by eating yogurt containing LAB with anti-inflammatory capacity.

## 2. Materials and Methods

### 2.1. Study Design

We performed a randomized, double-blind, placebo-controlled, parallel-group trial at a single site in Minato-ku, Tokyo, Japan between July and December 2018 (for participant recruitment and follow-up). The trial was conducted in accordance with the principles of the Declaration of Helsinki, and the protocol was approved by both the Ethical Committee of Shinagawa Season Terrace Health Care Clinic and the Meiji Institutional Review Board. All participants provided written informed consent before the screening test began. The study protocol was registered with the University Hospital Medical Information Network (UMIN) Clinical Trials Registry (UMIN000032837) on 1 June, 2018. This article completely conforms to the Consolidated Standards of Reporting Trials 2010 rules [19] (Appendix A). The data were analyzed between February and March 2019. We confirmed the accuracy of all analyzed data by self-inspection.

### 2.2. Participants

Study participants were recruited from a volunteer database associated with a contract research organization and were eligible to participate if they met the following inclusion criteria: In the screening test, they had to have fasting blood glucose (FBG) levels of 100–125 mg/dL and HbA1c levels of 5.6–6.4%. Additionally, they had to be otherwise healthy adults between 20 and 64 years of age. Candidates were excluded if they (1) took medication affecting blood glucose levels; (2) had a habit of taking supplements or health foods that affect blood glucose levels in the month prior to the screening test; (3) had a habit of drinking fermented milk or lactic fermenting beverages more than twice a week in the 3 months prior to the screening test; (4) had a milk allergy; (5) were diagnosed with diabetes as a result of the screening test, thus requiring medication; (6) had severe systemic diseases; (7) had a chronic illness which required them to take medication every day; (8) donated more than 200 mL of blood within a month or more than 400 mL within 3 months; (9) were heavy drinkers; (10) had drug or alcohol addictions; (11) participated in other clinical trials within the month prior to agreeing to be a participant in this study; (12) were pregnant or nursing; or (13) were judged as unsuitable for the study by the principal investigator for other reasons.

The target number of participants for enrollment was 130 (i.e., 65 participants per group), which enabled us to detect a group difference for HbA1c or FBG with a significance level of 5% and a power of 80%. The sample size was calculated based on the result of a preliminary clinical trial targeting prediabetic participants [17]. Considering that some candidates would refuse to participate in the trial and that many candidates would fail to meet the inclusion or meet exclusion criteria, the number of screening individuals had to be more than 1500.

### 2.3. Test Foods

The test foods were 112-g placebo yogurt or yogurt containing heat-treated OLL2712 (more than 5 × 10^9^ cells/112 g of yogurt). The assigned individual dispatched the test foods to each participant based on the group allocation. It was delivered to their home weekly under refrigeration and then kept refrigerated until consumption. The OLL2712 yogurt was confirmed to be identical in terms of appearance and flavor to the placebo yogurt. They both consisted of 58% dairy products, 3.4% sugar, 0.0038% sucralose, and 3% culture of yogurt starters and contained 76 kilocalories, 4.5 g proteins, 1.6 g fat, 10.8 g carbohydrates, 0.14 g sodium, and 0.12 g calcium. Heat-treated OLL2712 cells were prepared as previously described [17]. Briefly, the cells were grown at 33 °C to the late log-phase in a culture medium mainly composed of skim milk concentrate supplemented with mono-oleic acid esters with an initial pH of 6.6. During fermentation, the pH value was controlled with an automatic pH controller by the addition of 10% (*w*/*w*) KOH, and the pH was not allowed to decrease below 5.8. The cells after growth were heat treated at 60 °C for 10 min.

### 2.4. Randomization and Blinding

The participants were randomly assigned in a 1:1 ratio to receive either OLL2712 yogurt or placebo by a study staff who was not involved in the planning, enrollment, evaluation, intervention, or analysis of the trial by using a computer-generated random sequence. The participants were randomized in blocks of four by sex, age, HbA1c, FBG, and the homeostasis model assessment of insulin resistance (HOMA-IR) based on information obtained during the screening period to ensure a balance between these parameters in each group. All participants, investigators, and outcome assessors remained blinded to the group allocation until data analysis was completed.

### 2.5. Interventions

The study period consisted of a 4-week screening, followed by a 12-week treatment period. Each participant underwent medical interviews conducted by the physician in charge and had blood drawn five times in total: at the screening test and at the 0-, 4-, 8-, and 12-week tests.

Participants were instructed to ingest the 112-g test yogurt every day for 12 weeks (from the 0-week visit to the 12-week visit) and to visit the clinic every 4 weeks. For the study duration, participants were asked to maintain their normal diet and lifestyle habits including the quality and quantity of exercises. A life diary was provided to participants to confirm compliance and to report any problems. Participants were also asked to describe the type and amount of exercises they performed daily in their life diaries. The physician in charge monitored the participants’ compliance based on interviews and the diary entries at each clinic visit.

### 2.6. Diet Records

To exactly assess the efficacy of the yogurt, dietary intake was measured every 4 weeks. Before the intervention was started, participants received detailed written and verbal instructions on how to complete diet records. Participants recorded the details of meal contents in questionnaires to calculate the macronutrient intake for 3 days, including two weekdays and one weekend day, before each clinical examination. Food quantities were measured using standard measuring cups, spoons, and digital scales. The habitual nutrition intake and diet composition were assessed on the diet data analyzed with Excel Eiyokun ver. 8.0 (Kenpakusha, Tokyo, Japan).

### 2.7. Measurements

The primary outcome measures were changes in HbA1c and FBG levels. The secondary outcomes comprised changes in glucose metabolism-related parameters including glycoalbumin (GA), HOMA-IR, and chronic inflammation markers measured using blood, such as hsCRP and pro-inflammatory cytokine levels.

Body weight was measured by using a multi-frequency bioelectrical impedance device (InBody 430; Biospace, Seoul, Korea). Systolic and diastolic blood pressure readings were measured with a mechanized sphygmomanometer (Digital Automatic Blood Pressure Monitor HEM-907; Omron Healthcare, Kyoto, Japan).

Following overnight (≥12 h) fasting, a blood sample was drawn from the antecubital vein of each participant. The following blood parameters were analyzed by LSI Medicine (Tokyo, Japan): serum glucose, serum insulin, whole blood HbA1c, serum GA, and serum hsCRP. Serum pro-inflammatory cytokines levels were measured using a multiplex human cytokine bead array system (Bio-Rad, Hercules, CA, USA) or MESO QuickPlex SQ120 (Meso Scale Discovery, Rockville, MA, USA). Serum adiponectin levels were measured using the adiponectin enzyme-linked immunosorbent assay (ELISA) kit (ALPCO, Salem, NH, USA). HOMA-IR was calculated according to Equation (1) [20]:*HOMA-IR = fasting glucose (mg/dL) × fasting insulin (μU/mL)/*405(1)

### 2.8. Safety Assessment

At every clinic visit, the physician in charge checked the health status of the participants by means of a medical interview and in consultation with their diary entries. Hematology tests and blood biochemical tests were performed at the 0- and 12-week visits. Data on adverse events were obtained through the participants’ diary entries and the blood tests. The physician then judged whether a given unfavorable event was serious or nonserious and if it was study related.

### 2.9. Statistical Analysis

All data analyses were performed using IBM SPSS Statistics Ver. 24 (IBM Japan, Tokyo, Japan). *p*-values < 0.05 were considered statistically significant. The data were presented as means and standard deviations or standard errors. Unpaired *t*-tests or Mann–Whitney U tests were used to compare the intergroup changes, and Dunnett’s tests or Wilcoxon signed-rank tests were used to compare the intragroup change in the primary and secondary outcomes. Subgroup analyses were conducted for those meeting inclusion criteria at week 0 (baseline) as well as at the screening test and those with HOMA-IR levels of more than 1.6 at baseline, which is the upper limit of the normal HOMA-IR range in Japan [21]. In the safety assessment, the incidence of adverse events between the groups was analyzed using the Fisher’s exact test, and the values of the blood tests were compared between the groups by using unpaired *t*-tests.

## 3. Results

### 3.1. Participant Characteristics

The flowchart of participant’s entry into the study is shown in Figure 1. A total of 1593 individuals from the volunteer bank received a screening test at the study clinic and provided written informed consent. The participants were recruited from those who personally thought that they had high blood glucose levels. However, 1333 participants were excluded in accordance with the inclusion and exclusion criteria, and 21 declined to participate owing to personal reasons. Among the remaining 239 participants, 130 were selected in descending order, first based on their FBG levels and then based on their HbA1c levels, except for those who were judged to have any disease based on clinical examinations. They were enrolled in this study and assigned randomly to the OLL2712 (*n* = 65) or the placebo (*n* = 65) groups.

Of the 130 participants, 129 completed the 12-week intervention period while 1 declined because of personal circumstances. Adherence rate of the test food intake was more than 95%. After the exclusion of three additional participants (two were found to be ineligible after the 12-week visit, and one seriously violated the study compliance requirements) as determined by the principal investigator, 126 participants were included in the per-protocol-set (PPS) analysis. The characteristics of the participants measured at baseline and after the 12-week intervention are presented in Table 1. There were no significant differences in any characteristics between the two groups at week 0 and week 12, while body weight and blood pressure levels were significantly increased in both groups and heart rate levels were significantly decreased only in the OLL2712 group compared to those at baseline. The categorization of participants’ ages was as follows: 0 in the 20s, 10 in the 30s (*n* = 6 in the placebo group and *n* = 4 in the OLL2712 group), 39 in the 40s (*n* = 16 in the placebo group and *n* = 23 in the OLL2712 group), 63 in the 50s (*n* = 33 in the placebo group and *n* = 30 in the OLL2712 group), and 14 in the 60s (*n* = 9 in the placebo group and *n* = 5 in the OLL2712 group). The majority of participants in this study were in their 40s or 50s.

Daily nutrition intakes are shown in Table 2. No significant inter- or intragroup differences were found in these parameters. The nutrition intake derived from the test yogurt was not included in this result. Diet composition and lifestyle habits were also similar among all participants throughout the test period, which were confirmed by diary entries and the medical interviews conducted by the physician.

### 3.2. Effect of OLL2712 Yogurt on Glucose Metabolism-Related Parameters

Table 3 shows the changes in parameters that are related to glucose metabolism including FBG, HbA1c, GA, fasting insulin, and HOMA-IR. When compared to levels at baseline (103.4 ± 7.9 mg/dL and 103.8 ± 7.8 mg/dL in the placebo and OLL2712 groups, respectively), the FBG levels did not change significantly in both groups. However, the HbA1c levels were significantly reduced in both groups at week 12 compared to baseline (5.85 ± 0.21% and 5.86 ± 0.22% in the placebo and OLL2712 groups, respectively). The 12-week reduction of HbA1c levels was significantly greater in the OLL2712 group than in the placebo group (−0.07 ± 0.14% in the placebo group vs. −0.12 ± 0.14% in the OLL2712 group, *p* = 0.047; Figure 2). Fasting insulin levels were significantly increased in both groups compared to baseline; however, they continued to increase consistently throughout the study only in the placebo group. HOMA-IR levels were higher at week 12 than at baseline in the placebo group but not in the OLL2712 group. The GA levels at week 12 were lower in both groups than those at baseline. Overall, the only significant difference between the groups was found for HbA1c levels.

### 3.3. Effect of OLL2712 Yogurt on Chronic Inflammation

Table 4 shows the changes in serum pro-inflammatory cytokine levels, hsCRP levels, and adiponectin levels. Serum interleukin (IL)-6, IL-8, and hsCRP levels at week 12 were significantly increased compared to those at baseline in the placebo group but not in the OLL2712 group. Serum monocyte chemotactic protein-1 (MCP-1) levels at week 12 was significantly reduced compared to those at baseline in the OLL2712 group but not in the placebo group. Serum tumor necrosis factor (TNF)-α levels did not change significantly throughout the study. The adiponectin levels at week 12 were significantly increased in both groups compared to those at baseline.

### 3.4. Subgroup Analyses

Many participants with FBG levels within the normal range were observed at week 0 test; thus, we performed a subgroup analysis to eliminate the influence of these participants. The data on the participants who met the selection criteria both at the screening and at week 0 tests are shown in Table 5 (*n* = 40 and *n* = 43 in the placebo and OLL2712 groups, respectively; FBG levels of 107.0 ± 5.6 mg/dL and 107.6 ± 5.6 mg/dL in the placebo and OLL2712 groups, respectively). The FBG levels were significantly reduced at weeks 4 and 12 in the OLL2712 group compared to baseline but not in the placebo group. The GA levels were lower at week 12 in both groups than at baseline. In addition, fasting insulin levels were significantly increased at weeks 4 and 12 in the placebo group compared to baseline but not in the OLL2712 group. HbA1c levels at week 12 were significantly reduced in both groups compared to those at baseline. The 12-week change in HbA1c levels tended to decrease in the OLL2712 group compared to that in the placebo group (*p* = 0.093), although no significant difference was found between the groups due to insufficient number of participants.

Furthermore, we performed a subgroup analysis to determine if the ingestion of OLL2712 cells was particularly effective for participants with insulin resistance. The data for participants with HOMA-IR more than 1.6 at baseline are shown in Table 6 (*n* = 21 and *n* = 17 in the placebo and OLL2712 groups, respectively; FBG levels were 107.1 ± 8.2 mg/dL and 104.7 ± 8.7 mg/dL in the placebo and OLL2712 groups, respectively). At week 12, the FBG levels tended to decrease in the OLL2712 group compared to those at baseline (*p* = 0.059) but not in the placebo group. Additionally, the blood GA levels significantly decreased only in the OLL2712 group at weeks 8 and 12 compared to baseline. The mean fasting insulin levels and HOMA-IR levels were increased consistently in the placebo group and were decreased consistently in the OLL2712 group compared to baseline although statistically significant differences were not observed.

### 3.5. Safety Assessment

No serious adverse events occurred in the study, although a total of 21 nonserious adverse events were accounted for through participants’ diaries, medical interviews, and blood tests. There were no significant differences in the incidence of adverse events between the groups, and none of these were judged by the physician in charge to be related to the consumption of the test yogurt. Regarding the blood tests, the physician judged that the changes in any parameter were within normal ranges.

## 4. Discussion

To investigate the preventive contribution of LAB with anti-inflammatory capacity against the onset of T2DM, we targeted prediabetic participants and found that the supplementation with heat-treated OLL2712 cells for 12 weeks reduced the HbA1c levels of the participants. HbA1c levels were significantly lower in both groups at week 12 than at baseline, but the decrease was significantly greater in the OLL2712 group than in the placebo group. The important point was that the decline of HbA1c levels occurred without lifestyle intervention such as caloric restriction or increased physical activity. The average values of total energy and carbohydrate intake increased slightly at week 12 compared to baseline in both groups, but there was no significant difference between and within groups. Therefore, the effect of diet on the results of this study is considered minor (Table 2). Food restriction or forced exercise are difficult to continue and do not always secure long-term benefits for all people [7,8]. We used a 112-g yogurt as a test food, which can be consumed daily for long periods. The reduction in the HbA1c levels reported in this study was not very different from that reported in another study in which participants received a 36-week intervention including nutritional counseling by dietitians [22]. Therefore, although the effect size was very small compared to that of clinical trials with the use of antidiabetic medication that target patients with severe diabetes, it was worthwhile to show the statistically significant reduction in HbA1c levels as a result of the consumption of food ingredients.

It is necessary to consider the seasonal variation in HbA1c levels when interpreting the results of this study. Test foods were ingested from summer to winter (from August to November) in this study. Generally, in the winter season, it is known that body weight increases and that blood glucose and HbA1c levels also increase compared to those in the summer season [23,24,25,26]. Thus, we consider that the reason of increased body weight of the participants in this study was probably due to this seasonal variation (Table 1). Over such a period, we showed that HbA1c levels at week 12 were significantly decreased compared to those at baseline, not only in the OLL2712 group but also in the placebo group (Table 3).

Yogurt intake has been reported to reduce the risk of developing T2DM [27,28,29]. A recent meta-analysis by Gijsbers et al. also reported an inverse association between yogurt consumption and T2DM risk among various dairy products investigated [30]. Yogurt consumption maintains the gut barrier function by controlling the intestinal microbiome and its metabolites [31,32]. Repairing damaged mucus layers and tight junctions of the intestine may prevent “leaky gut” and may avoid the onset of diabetes [33,34]. In addition, peptides derived from milk proteins were reported to induce incretin including GLP-1 [35]. Yogurt consumption is expected to have an effect of improving glucose metabolism through improvement in gut barrier function and the induction of GLP-1. Therefore, although yogurt decreased the HbA1c levels during the period when body weight and HbA1c are likely to increase, it was considered that the addition of OLL2712 cells further enhanced this reduction.

On the other hand, no significant reduction in FBG levels was observed on both intra- and intergroup comparisons despite the reduction in the HbA1c levels in the PPS analysis (Table 3). The screening test was conducted for more than 1500 candidates to ascertain enough participants with FBG levels from 100 to 125 mg/dL, and the baseline (week 0) test was carried out approximately one month after the screening test. Consequently, for one-third of the participants, the FBG levels dropped below 100 mg/dL at baseline because FBG levels fluctuate easily and are affected by the last meals. Many participants in this study exhibited low baseline levels of FBG, and no major effects on FBG levels by ingesting OLL2712 cells can be expected in these participants, which might be the probable reason that OLL2712 cells did not decrease the FBG levels in the PPS analysis. Assuming this case, we performed subgroup analyses to exclude the influence of participants with lower FBG levels. In the subgroup analysis of participants with FBG levels higher than the upper range of normal, significant decreases in FBG levels after 12 weeks were observed when compared to the baseline in the OLL2712 group but not in the placebo group (Table 5). In the subgroup analysis of participants with HOMA-IR levels higher than the upper range of normal, FBG levels tended to decrease and GA levels were significantly decreased at week 12 compared to baseline in the OLL2712 group but not in the placebo group (Table 6). In contrast, HbA1c level is a stable marker reflecting not only FBG levels but also postprandial blood glucose levels. More than 95% of the participants met the inclusion criteria of HbA1c also at the week 0 test. It was suggested that HbA1c levels are closely associated with postprandial glucose levels in comparison with FBG levels [36]. Postprandial glucose excursions contribute more to HbA1c in participants with lower FBG levels [37,38]. It is known that, among those with low FBG levels, there are people whose postprandial blood glucose level will sharply increase. Therefore, the ingestion of OLL2712 cells might reduce the HbA1c levels in participants with lower FBG levels by suppressing postprandial glucose excursions. Given the above, the ingestion of OLL2712 cells may be able to control the glycemic levels.

The major cause of glucose metabolism deterioration is considered chronic inflammation and insulin resistance [9]. Numerous studies have investigated the correlation between blood pro-inflammatory cytokine levels and metabolic disorders [9,39,40]. IL-6 secreted by T cells, macrophages, and adipocytes plays a key role in the development of insulin resistance and atherosclerosis, pathologies related to obesity and metabolic syndrome [41]. IL-6 promotes the production by T cells and macrophages of CRP associated with increased risk of diabetes, hypertension, and cardiovascular disease [9]. Chronic inflammation is known to inhibit insulin signaling and to impair GLUT4 translocation to the cell membrane, thereby inducing insulin resistance [42,43]. Therefore, the occurrence of chronic inflammation is thought to cause insulin resistance, to worsen glucose metabolism, and to lead to an increase in HbA1c levels. In our previous studies, we demonstrated that the administration of OLL2712 cells significantly improved insulin resistance [15] and significantly suppressed pro-inflammatory cytokine levels in visceral adipose tissue and the serum [14,15,16] in mice models with obesity and T2DM. We also reported that the administration of OLL2712 cells significantly improved HOMA-IR levels compared to those at baseline and that the improvement in HOMA-IR levels was prominent in participants with chronic inflammation at baseline [17]. These results showed that OLL2712 cells exhibit high anti-inflammatory activity and are characterized by their ability to suppress chronic inflammation and insulin resistance.

In the present study, blood IL-6, IL-8, and hsCRP levels as well as HOMA-IR levels significantly increased in the placebo group at week 12 compared to baseline but not in the OLL2712 group (Table 3 and Table 4). In subgroup analyses, the mean fasting insulin levels and HOMA-IR levels increased in the placebo group and decreased in the OLL2712 group (Table 5 and Table 6). In both groups, the participants gained weight, and fasting insulin levels were likely increased due to seasonal fluctuations from summer to winter, suggesting that the deterioration of insulin resistance and glucose tolerance was suppressed by ingesting OLL2712 cells. The supplementation with OLL2712 cells might reduce HbA1c levels via the suppression of the aggravation of chronic inflammation and insulin resistance. However, there were no significant differences between the groups based on these parameters and their causality was unknown in this study. One of the reasons for this may be that the participants in this study often had relatively low levels of chronic inflammation; the average serum IL-6 level was less than 1 pg/mL. The findings from our preliminary study suggested that the effect of OLL2712 cells was more pronounced on participants with high levels of chronic inflammation whose average serum IL-6 level was more than 2 pg/mL [17]. Thus, future studies are needed to clarify the mechanism of action of OLL2712 cells. We suggest that the effects of OLL2712 cells may be more prominent in participants with higher levels of chronic inflammation and insulin resistance.

There are a limited number of clinical trials that have assessed the efficacy of LAB for reducing HbA1c levels. In particular, the benefit of the addition of specific LAB strains for the prevention of T2DM compared to conventional yogurt has not been clarified. Recently, Barengolts et al. reviewed nine randomized controlled trials (RCTs) that utilized probiotic yogurt as the main intervention in participants with T2DM or obesity using meta-analysis. There was no significant difference in the pooled unstandardized mean difference for HbA1c levels based on the use of probiotic yogurts versus conventional yogurt [4]. In most of these trials, however, the intake periods were short such as 6 or 8 weeks and the participants were T2DM patients. It is preferable that the intake period be 12 weeks or greater to assess the impact of probiotic supplementation on HbA1c because HbA1c levels reflect the average blood glucose levels in the past 1 to 2 months. In addition, prediabetic participants are suitable for verifying the preventive effect of functional food such as yogurt because medication is usually prioritized for diabetic patients. To our knowledge, we are the first investigators to demonstrate that yogurt containing anti-inflammatory LAB cells is effective in reducing the HbA1c levels in prediabetic participants. Additionally, these findings suggest that anti-inflammatory LAB cells have the potential to prevent T2DM by suppressing the aggravation of chronic inflammation.

The limitation of this study was the absence of a control group that consumed zero-calorie soft drink or sour milk rather than yogurt. The placebo yogurt used in this study contained some effective ingredients including more than 10^11^ cells of *Lactobacillus bulgaricus* and *Streptococcus thermophilus*, which might provide glycemic improvement and might affect the benefits of OLL2712 cells. As it was difficult to use a placebo in this double-blind manner to rigorously test the efficacy of yogurt intake, only conventional yogurt was used as the control to examine the further effect of the addition of OLL2712 cells. In this study, we wanted to confirm that, when specific LAB cells with high anti-inflammatory activity were added to yogurt, the effects of yogurt on improving glucose metabolism would be further enhanced without negating each other’s effects. We would like to contribute to the creation of foods that are easy to eat and that produce beneficial effects on improving glucose metabolism. Another limitation was that we did not analyze fecal microbiota or metabolites by sampling feces of the participants. The intestinal microbiota has been associated with the development of various human diseases including metabolic syndrome, obesity, and related metabolic dysfunctions [44]. Although OLL2712 cells were heat treated, the possibility of affecting intestinal microbiota cannot be denied. In addition, we did not confirm whether heat-treated OLL2712 cells reached the intestine of the participants. Lastly, oral glucose tolerance test (OGTT) is necessary to examine the effect of OLL2712 ingestion on postprandial blood glucose levels; however, we could not conduct OGTT in the present study. To further clarify the improvement effect of OLL2712 cells on glucose metabolism in humans, OGTT is a useful method. In future clinical studies, we will perform these additional analyses to clarify the beneficial effects and the mechanism of action of OLL2712 cells.

## 5. Conclusions

The ingestion of OLL2712 cells reduced the HbA1c levels in prediabetic adults. The action mechanism of OLL2712 cells to control blood glucose levels might be the prevention of the aggravation of chronic inflammation and insulin resistance.

## Figures and Tables

**Figure 1 nutrients-12-00374-f001:**
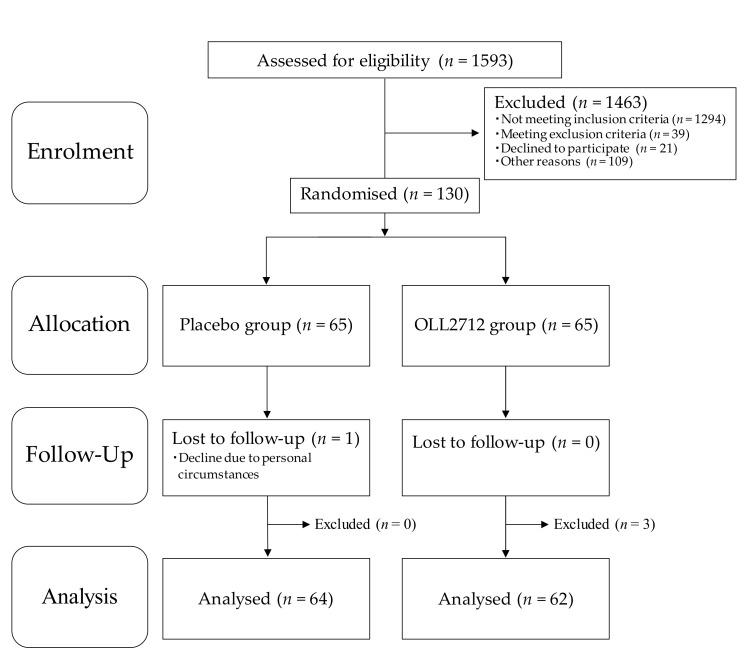
Flowchart of study participants: Placebo, conventional yogurt; OLL2712, yogurt containing more than 5 × 10^9^ heat-treated *Lactobacillus plantarum* OLL2712 cells.

**Figure 2 nutrients-12-00374-f002:**
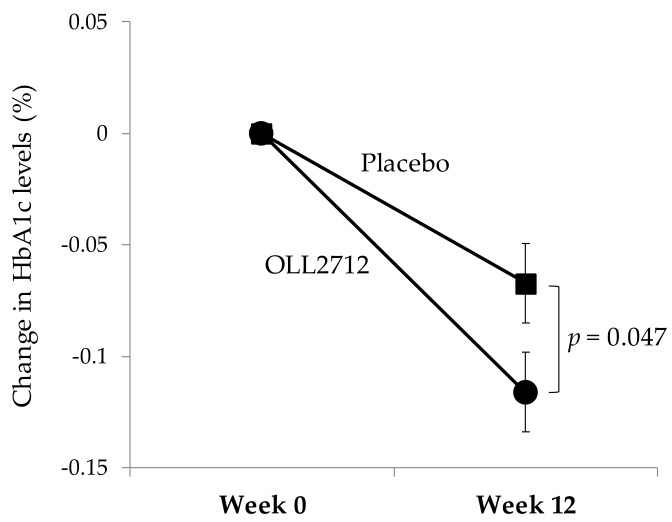
The 12-week change in HbA1c levels in each test group: Each data point represents the mean value (*n* = 64 and *n* = 62 in the placebo and OLL2712 groups, respectively). Error bars indicate standard errors.

**Table 1 nutrients-12-00374-t001:** Baseline characteristics of study participants and changes in these characteristics after the 12-week intervention.

	Placebo Yogurt (*n* = 64)		OLL2712 Yogurt (*n* = 62)	
Characteristics	Week 0	Week 12	Week 0	Week 12
Age (years)	51.2 ± 7.6	NA	50.6 ± 6.9	NA
Male/Female	44/20	NA	42/20	NA
BW (kg)	69.4 ± 12.3	70.3 ± 12.5 **	69.1 ± 11.0	69.8 ± 10.8 **
BMI (kg/m^2^)	24.9 ± 3.2	25.2 ± 3.4 **	24.7 ± 3.3	25.0 ± 3.2 **
SBP (mmHg)	123.3 ± 14.3	128.3 ± 14.2 **	126.6 ± 14.4	133.1 ± 13.2 **
DBP (mmHg)	74.4 ± 10.7	79.7 ± 9.7 **	77.4 ± 11.0	82.0 ± 11.2 **
HR (beats/min)	76.4 ± 11.4	76.7 ± 12.1	76.7 ± 12.2	73.8 ± 12.2 *

Abbreviations: BW, body weight; BMI, body mass index; SBP, systolic blood pressure; DBP, diastolic blood pressure; HR, heart rate; NA, not assessed. Data are presented as means ± standard deviations. Significant differences in measurements compared to those at week 0 (baseline) were determined using the paired *t*-tests or the Wilcoxon signed-rank tests (* *p* < 0.05, ** *p* < 0.01). Significant differences in measurements compared to the placebo group were determined using the unpaired *t*-test or the Mann–Whitney U tests, and there were no significant differences.

**Table 2 nutrients-12-00374-t002:** Daily nutrition intakes.

Variable	Group	Week 0	Week 4	Week 8	Week 12
Total energy (kcal/day)	Placebo	1817 ± 414	1831 ± 500	1860 ± 450	1854 ± 497
	OLL2712	1838 ± 355	1775 ± 404	1809 ± 407	1857 ± 376
Protein (g/day)	Placebo	69.0 ± 20.7	69.2 ± 20.3	68.9 ± 18.3	70.2 ± 20.0
	OLL2712	67.2 ± 15.8	65.5 ± 14.9	67.3 ± 17.9	68.5 ± 18.9
Fat (g/day)	Placebo	62.6 ± 20.2	63.3 ± 22.8	63.9 ± 19.9	64.5 ± 23.8
	OLL2712	65.8 ± 18.0	62.9 ± 19.0	63.0 ± 21.7	65.9 ± 18.9
Carbohydrate (g/day)	Placebo	232 ± 58	233 ± 72	239 ± 66	236 ± 76
	OLL2712	230 ± 51	223 ± 59	229 ± 52	234 ± 52
Dietary fiber (g/day)	Placebo	11.0 ± 3.3	11.1 ± 4.0	11.3 ± 3.8	11.3 ± 4.0
	OLL2712	10.8 ± 3.1	10.6 ± 3.3	11.0 ± 3.1	11.3 ± 2.5

Participants recorded the details of meal contents in questionnaires to calculate the macronutrient intake for 3 days before each clinical examination. The habitual nutrition intake and diet composition were assessed on the diet data analyzed with Excel Eiyokun ver. 8.0. Data are presented as means ± standard deviations. Significant differences in measurements compared to those at week 0 (baseline) were determined using the Dunnett’s tests or the Wilcoxon signed-rank tests; significant differences in measurements compared to that of the placebo group were determined using the unpaired Student’s *t*-test or the Mann–Whitney U test, and there were no significant differences.

**Table 3 nutrients-12-00374-t003:** Blood biochemical parameters related to glucose metabolism and insulin resistance.

Variable	Group	Week 0	Week 4	Week 8	Week 12	12-Week Change
FBG (mg/dL)	Placebo	103.4 ± 7.9	101.0 ± 8.9	104.6 ± 9.6	101.4 ± 11.8	−2.0 ± 10.8
	OLL2712	103.8 ± 7.8	102.3 ± 9.7	103.6 ± 10.3	102.1 ± 9.5	−1.7 ± 8.1
HbA1c (%)	Placebo	5.85 ± 0.21	5.79 ± 0.22^**^	5.89 ± 0.22 **	5.78 ± 0.24 **	−0.07 ± 0.14
	OLL2712	5.86 ± 0.22	5.82 ± 0.25^*^	5.89 ± 0.26	5.74 ± 0.26 **	−0.12 ± 0.14 ^#^
GA (%)	Placebo	14.8 ± 1.2	14.7 ± 1.2	14.5 ± 1.2 **	14.5 ± 1.2 **	−0.2 ± 0.5
	OLL2712	14.6 ± 1.1	14.5 ± 1.1	14.3 ± 1.2 **	14.2 ± 1.1 **	−0.3 ± 0.5
Insulin (μU/mL)	Placebo	5.64 ± 2.87	6.40 ± 3.16 *	6.42 ± 3.28 **	7.31 ± 5.03 **	1.68 ± 4.03
	OLL2712	5.88 ± 4.78	5.82 ± 2.88 *	6.16 ± 3.08 **	6.07 ± 3.24 *	0.24 ± 4.41
HOMA-IR	Placebo	1.45 ± 0.82	1.60 ± 0.82	1.68 ± 0.95 **	1.87 ± 1.38 *	0.42 ± 1.12
	OLL2712	1.52 ± 1.32	1.48 ± 0.76	1.58 ± 0.79*	1.53 ± 0.80	0.02 ± 1.25

Abbreviations: FBG, fasting blood glucose; GA, glycoalbumin; HbA1c, whole blood hemoglobin A1c; HOMA-IR, homeostasis model assessment of insulin resistance. Data are presented as means ± standard deviations. Significant differences in measurements compared to those at week 0 (baseline) were determined using the Dunnett’s tests or the Wilcoxon signed-rank tests (* *p* < 0.05, ** *p* < 0.01). Significant differences in measurements compared to those in the placebo group were determined using the unpaired *t*-test or the Mann–Whitney U test (^#^
*p* < 0.05).

**Table 4 nutrients-12-00374-t004:** Serum pro-inflammatory cytokines, high-sensitivity C-reactive protein, and adiponectin.

Variable	Group	Week 0	Week 4	Week 8	Week 12	12-Week Change
IL-6 (pg/mL)	Placebo	0.532 ± 0.268	0.496 ± 0.250	0.563 ± 0.310	0.601 ± 0.265 *	0.070 ± 0.236
	OLL2712	0.801 ± 1.231	0.855 ± 1.456	0.779 ± 1.453	0.792 ± 1.296	−0.001 ± 0.507
IL-8 (pg/mL)	Placebo	4.71 ± 2.52	2.65 ± 1.81 **	6.21 ± 5.80 *	5.72 ± 3.46 *	1.01 ± 3.12
	OLL2712	5.24 ± 2.11	2.85 ± 1.76 **	5.41 ± 2.23	6.66 ± 8.41	1.38 ± 8.56
MCP-1 (pg/mL)	Placebo	26.8 ± 21.7	16.6 ± 7.8 **	21.2 ± 11.1 **	24.9 ± 17.2	−1.95 ± 24.53
	OLL2712	25.3 ± 11.9	16.6 ± 9.1 **	21.2 ± 10.8 **	23.5 ± 11.1 *	−2.10 ± 8.87
TNF-α (pg/mL)	Placebo	9.79 ± 5.22	6.79 ± 8.36	11.04 ± 5.41	10.09 ± 4.69	0.29 ± 3.64
	OLL2712	9.93 ± 4.37	5.17 ± 3.34	10.99 ± 4.84	10.23 ± 5.41	0.20 ± 4.62
hsCRP (mg/dL)	Placebo	0.058 ± 0.067	NA	NA	0.079 ± 0.098 **	0.030 ± 0.078
	OLL2712	0.048 ± 0.049	NA	NA	0.054 ± 0.049	0.006 ± 0.031
Adiponectin (μg/mL)	Placebo	5038 ± 1924	5202 ± 2275	5233 ± 2090	5478 ± 2156 **	439 ± 780
	OLL2712	5472 ± 2002	5468 ± 2136	5559 ± 1975	6047 ± 2185 **	576 ± 1078

Abbreviations: IL, interleukin; MCP-1, monocyte chemotactic protein-1; TNF-α, tumor necrosis factor-α; hsCRP, high-sensitivity C-reactive protein; NA, not assessed. Data are presented as means ± standard deviations. Significant differences in measurements compared to those at week 0 (baseline) were determined using the Dunnett’s tests or the Wilcoxon signed-rank tests (* *p* < 0.05, ** *p* < 0.01). Significant differences in measurements compared to those in the placebo group were determined using the unpaired *t*-test or the Mann–Whitney U test, and there were no significant differences.

**Table 5 nutrients-12-00374-t005:** Subgroup analysis in the participants who met the selection criteria at week 0 test.

Variable	Group	Week 0	Week 4	Week 8	Week 12	12-Week Change
FBG (mg/dL)	Placebo	107.0 ± 5.6	104.1 ± 8.2	108.1 ± 8.6	104.9 ± 10.4	−2.2 ± 10.4
	OLL2712	107.6 ± 5.6	104.9 ± 9.4^*^	107.1 ± 9.4	104.8 ± 8.7 *	−2.8 ± 8.5
HbA1c (%)	Placebo	5.92 ± 0.21	5.87 ± 0.21^**^	5.98 ± 0.21 **	5.85 ± 0.25 **	−0.07 ± 0.16
	OLL2712	5.90 ± 0.22	5.87 ± 0.24	5.94 ± 0.24	5.77 ± 0.26 **	−0.12 ± 0.14
GA (%)	Placebo	14.9 ± 1.2	14.9 ± 1.2	14.7 ± 1.1 *	14.7 ± 1.2 *	−0.2 ± 0.5
	OLL2712	14.7 ± 1.2	14.7 ± 1.2	14.4 ± 1.2 **	14.3 ± 1.1 **	−0.3 ± 0.5
Insulin (μU/mL)	Placebo	5.70 ± 2.47	6.43 ± 2.98 *	6.51 ± 3.65	7.34 ± 4.77 *	1.64 ± 3.90
	OLL2712	6.24 ± 5.28	5.82 ± 2.92	6.19 ± 2.67	5.87 ± 2.67	−0.37 ± 4.89
HOMA-IR	Placebo	1.51 ± 0.66	1.66 ± 0.78	1.77 ± 1.09	1.95 ± 1.39	0.44 ± 1.13
	OLL2712	1.67 ± 1.48	1.52 ± 0.80	1.65 ± 0.75	1.52 ± 0.72	−0.15 ± 1.41

Abbreviations: FBG, fasting blood glucose; GA, glycoalbumin; HbA1c, whole blood hemoglobin A1c; HOMA-IR, homeostasis model assessment of insulin resistance. Data are presented as means ± standard deviations (*n* = 40 and *n* = 43 in the placebo and OLL2712 groups). Significant differences in the measurements compared to those at week 0 (baseline) were determined using the Dunnett’s tests or the Wilcoxon signed-rank tests (* *p* < 0.05, ** *p* < 0.01). Significant differences in the measurements compared to those in the placebo group were determined using the unpaired *t*-test or the Mann–Whitney U test, and there were no significant differences.

**Table 6 nutrients-12-00374-t006:** Subgroup analysis in the participants whose HOMA-IR levels were more than 1.6 at week 0 test.

Variable	Group	Week 0	Week 4	Week 8	Week 12	12-Week Change
FBG (mg/dL)	Placebo	107.1 ± 8.2	103.9 ± 8.1	109.3 ± 10.9	105.5 ± 10.2	−1.6 ± 10.5
	OLL2712	104.7 ± 8.7	102.6 ± 8.4	103.7 ± 11.7	100.0 ± 11.4	−4.7 ± 10.6
HbA1c (%)	Placebo	5.94 ± 0.18	5.88 ± 0.20	6.01 ± 0.21	5.90 ± 0.26	−0.03 ± 0.17
	OLL2712	5.94 ± 0.23	5.92 ± 0.23	5.99 ± 0.24	5.89 ± 0.29	−0.05 ± 0.17
GA (%)	Placebo	14.6 ± 1.5	14.6 ± 1.5	14.5 ± 1.4	14.5 ± 1.5	−0.1 ± 0.5
	OLL2712	14.4 ± 1.1	14.3 ± 1.0	14.1 ± 1.0 **	14.2 ± 1.1 *	−0.3 ± 0.5
Insulin (μU/mL)	Placebo	8.61 ± 2.88	9.13 ± 2.66	9.38 ± 3.30	10.88 ± 5.03	2.27 ± 5.30
	OLL2712	10.53 ± 6.93	8.40 ± 3.29	9.09 ± 3.15	8.68 ± 4.02	−1.85 ± 7.81
HOMA-IR	Placebo	2.23 ± 0.89	2.34 ± 0.70	2.57 ± 1.05	2.89 ± 1.51	0.60 ± 1.55
	OLL2712	2.74 ± 1.97	2.12 ± 0.84	2.32 ± 0.76	2.15 ± 0.97	−0.59 ± 2.22

Abbreviations: FBG, fasting blood glucose; GA, glycoalbumin; HbA1c, whole blood hemoglobin A1c; HOMA-IR, homeostasis model assessment of insulin resistance. Data are presented as mean ± SD (*n* = 21 in the placebo group and *n* = 17 in the OLL2712 group). Significant differences compared to measurements at week 0 (baseline) were determined by the Dunnett’s test or the Wilcoxon signed-rank test (* *p* < 0.05, ** *p* < 0.01). Significant differences compared to the placebo group were determined by the unpaired *t*-test or the Mann–Whitney U test, and there were no significant differences.

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
