# Peer review of "Effects of 12-Week Ingestion of Yogurt Containing Lactobacillus plantarum OLL2712 on Glucose Metabolism and Chronic Inflammation in Prediabetic Adults: A Randomized Placebo-Controlled Trial"

_nutrients, 2020, doi:10.3390/nu12020374_

Round 1
Reviewer 1 Report
thank you for revising your manuscript
Reviewer 2 Report
The study by Takayuki Toshimitsu et al. entitled “Effects of 12-week ingestion of yogurt containing Lactobacillus plantarum OLL2712 on glucose metabolism and chronic inflammation in pre-diabetic adults: A randomized placebo-controlled trial” has been reviewed. All suggestions and questions provided by the reviewers were promptly and carefully approved by the authors and correctly modified in the text. I would suggest a minor revision of the English language for the new parts inserted.
This manuscript is a resubmission of an earlier submission. The following is a list of the peer review reports and author responses from that submission.
Round 1
Reviewer 1 Report
This is an interesting study describing the effects of LAB OLL2712 on glucose metabolism and inflammation in pre-diabetic adults. With an increasing interest in functional foods these days, this is a timely research project not only for pre-diabetes but also for any chronic disease with an involvement of chronic inflammation e.g. cardiovascular disease.
General comments
Subjects and participants is used interchangeably - would be best to be consistent and use participants throughout the paper.
Tables should be formatted in a way that the results are on one line to avoid spacing issues.
Please review the discussion of results on HbA1c - according to the results HbA1c levels decreased in both groups.
Please review the discussion of the results for FBG and GA - according to the results table GA was also significantly reduced in the higher FBG placebo group, see Table 5.
There is no discussion about fasting insulin levels - this needs to be added.
Any changes will need to be incorporated into the abstract.
Edits
line 118 - was this one staff member? then it is 'a study staff who was not involved...' - if it was more than one, it should read 'by study staff who were not involved...'
line 182 - 'subjects' has a different font size
line 185 - 'subjects' has a different font size
line 186 - 'except for those who were presumed to have any disease' please reword to clarify
line 204 - insert a return before 'Data' similar to the formatting in other table legends
line 223 - specify that this is fasting insulin
line 225 - the HbA1c levels were .....
line 228 - insert results for the two groups and delete 'those'
line 229 - not only in the placebo group - fasting insulin levels were significantly increased in both groups
line 231 - you might want to add that the only significant difference between groups was found for HbA1c
line 243 - each data point is not n=126; more n=64 or n=62
line 245-251 - what about TNF-∝?
line 260 - section 3.4 - the results for GA and fasting insulin are also significant and this is not described.
line 269 - it might be good to add that no significant differences were found between treatment groups.
line 305 - you are leaving out that the reduction in HbA1c was highly significant for both groups and significant between treatment groups with the LAB group showing a significant reduction in HbA1c compared to the placebo group
line 340-343 - this statement is not true, at least not according to the data for GA. And what about fasting insulin levels?
line 368 - 'one of the reasons for this....'
line 372 - the action mechanism is usually called the mechanism of action
line 372-374 - suggest rewording this sentence to something like 'We suggest that the effects of OLL2712 cells may be more prominent in participants with higher levels of chronic inflammation and insulin resistance.'
line 390-396 - wondering why this was not done - use of a placebo which did not contain any potentially useful bacteria
line 398-399 - mechanism of action
Overall, this is an interesting study. It is well presented and written and certainly provides food for thought.
Reviewer 2 Report
Authors insist that yogurt containing Lactobacillus plantarum OLL2712 effect on glucose metabolism and chronic inflammation in pre-diabetic adults. I think this manuscript have several fundamental problems. My specific comments are following.
Authors analyzed data from 126 subjects (64 subjects for placebo yogurt, 62 subjects for OLL2712 yogurt). The age of subjects enrolled in the study is 20~64 years. The glucose metabolism and parameters of chronic inflammation deeply influence by age. For this reason, authors must consider variation on glucose metabolism and parameters of chronic inflammation by age. The HbA1c was significantly reduced in OLL2712 group than that of placebo group (Table 3). But, the FBG was not significantly difference between placebo group and OLL2712 group. What is reason of this? In the subgroup analysis (Table 5, 6), there is no significant difference on HbA1c between placebo group and OLL2712 group. This result is discordant with that of Table 3. How authors explain this discrepancy? To know improvement of glucose metabolism in human, oral glucose tolerance test (OGTT) is useful method. Authors should include OGTT data to improve quality of manuscript.
Reviewer 3 Report
A well-designed RCT by Toshimitsu et al, to test the efficacy of L.plantarum in pre-diabetics. The introduction is good but can be improved by expanding on similar previous studies and their interpretation and hence setting up a clear rationale.
The material and methods section is fine. In 2.3 test foods, I would like to see the complete content of the placebo.
Results Tables are done well. In Table 2, the authors present daily nutrition intakes. Can they explain the method? The discussion section should point to diet as a potential confounder and discuss how it might have impacted the results of this study.
Can the authors explain the rationale for systemic cytokine level measurements and correlating them as anti-inflammatory effects? What is the link between the microbiome and systemic inflammation? Can the authors also comment on how the probiotic strains have settled in the colon? was there any measurement of probiotic strain in stools for example?
The authors should expand on hoe the placebo containing beneficial bacteria was chosen and also their potential effect or lack of effect in those individuals?